# CALIBRATING THE VOICE OF DOUBT: HOW LLMS DIVERGE FROM HUMANS IN VERBAL UNCERTAINTY

## ABSTRACT

Humans naturally express uncertainty through verbal cues via uncertainty markers (e.g., "*possible*", "*likely*"), yet existing Large Language Model (LLM) uncertainty quantification (UQ) methods primarily rely on response likelihood or semantic consistency, which are often computationally costly. Despite increasing interest in LLM reliability, it remains underexplored how LLMs diverge from humans in verbal uncertainty expression: *Do LLMs share the same confidence level of uncertainty markers as humans? Can we quantify LLM uncertainty verbally?* To address this gap, we study the divergence between humans and LLMs in verbal uncertainty expression. Specifically, we first collect a corpus of human uncertainty markers from the literature and systematically examine their alignment with LLMs. Our extensive experiments reveal that LLMs may encode verbal uncertainty with confidence levels that differ substantially from those of humans. To bridge this mismatch, we introduce VOCAL, a novel optimization-based algorithm that learns the confidence level for each uncertainty marker for LLMs. VOCAL achieves comparable performance on par with state-of-the-art sampling-based UQ methods over extensive experimental settings, with significantly reduced computational costs. Moreover, VOCAL disentangles the calibration mismatch and pinpoints the confidence disparity between human and LLM verbal expressions. This work opens a new perspective on LLM UQ by grounding it in the verbal dimension of uncertainty expression, and offers insights into both model alignment and human–AI communication.

## 1 INTRODUCTION

Despite large language models' (LLMs) recent remarkable success across diverse domains (Yang et al., 2024; Thapa et al., 2025; Xie et al., 2023; Colombo et al., 2024), a fundamental question remains: when should we trust LLMs' responses? This question highlights the need to make LLMs more trustworthy and responsible. Hallucinations are not only mistakes but also risks that can reduce users' trust and cause harm in sensitive applications (Asgari et al., 2025; Das et al., 2025), like giving unsafe treatment advice in biomedicine. One promising approach to mitigating this phenomenon is uncertainty quantification (UQ) (Malinin & Gales, 2020; Kuhn et al., 2023a; Duan et al., 2024), which aims to measure and express the confidence of a model in its predictions. UQ provides a probabilistic signal of reliability directly from the model's outputs. This enables the estimation of a prediction's trustworthiness even in the absence of labeled data, a scenario common in real-world applications, and to distinguish between cases where the model is likely correct and those where it may be uncertain, extrapolating, or hallucinating.

However, existing approaches for quantifying hallucination in LLMs still have some limitations. Most current methods can be broadly divided into two main groups: sampling-based techniques (Farquhar et al., 2024; Kossen et al., 2024; Li et al., 2025; McCabe et al., 2025) and logits-based techniques (Nguyen et al., 2025; Ma et al., 2025; Yang et al., 2025; Sriramanan et al., 2024). Malinin & Gales (2020) introduced predictive entropy (PE), a logits-based technique that can give useful reliability estimates but often mistakes simple wording changes for uncertainty and usually requires heavy computation. Sampling-based methods, such as semantic entropy (SE) Kuhn et al. (2023b), ensemble variance, or consistency checks across multiple generations, can be more robust but are also slow and costly, which makes them difficult to use in practice. Therefore, a recent direction focuses on verbal uncertainty, where models are asked to output a confidence score (often on a 1–100 scale) in natural language form (Tian et al., 2023b). While this strategy can improve calibration

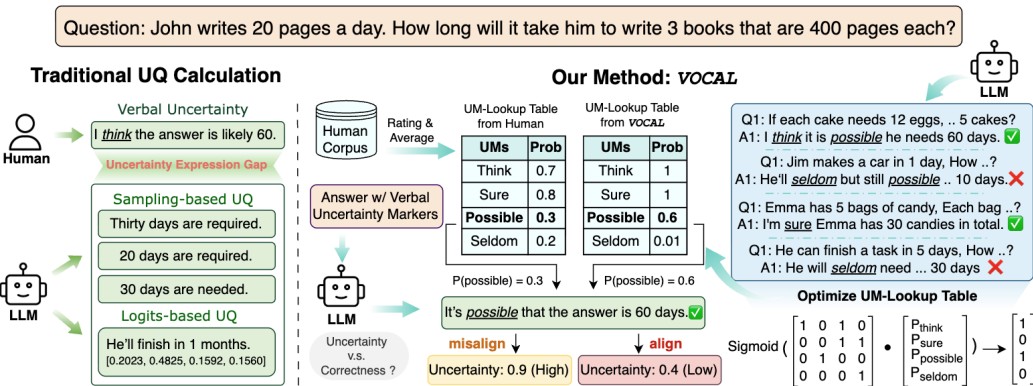

Figure 1: Comparison of traditional uncertainty quantification (UQ) methods and our method VOCAL. Traditional UQ methods (sampling-based and logits-based) exhibit a gap with human uncertainty expressions. In VOCAL, UM-lookup tables derived from human data alone cannot fully capture model uncertainty, so they are optimized with the model's confidence distribution to better align with its internal uncertainty expressions.

compared to raw probability outputs, it is still unnatural because humans do not usually express uncertainty as exact numbers. Instead, people prefer qualitative terms such as "*possible*," "*likely*," or "*almost certain*" in daily communication. These expressions are easier to understand and better capture the nuance of human reasoning. This contrast shows a gap in current methods and points to the need for approaches that allow models to express uncertainty in a way that is more natural, human-like, and trustworthy for real-world use (Figure 1 (left)).

Motivated by this gap, our work investigates whether LLMs can express verbal uncertainty in a manner comparable to humans. To study this question, we construct the first verbal uncertainty marker lookup table (UM-Lookup) that maps qualitative expressions of uncertainty to numerical representations. The lookup table is built through a literature review grounded in psychology and decision science (Lichtenstein & Newman, 1967; Beyth-Marom, 1982; Wesson & Pulford, 2009), followed by a debiasing procedure to refine ambiguous cases. We then aggregate judgments from more than 300 human annotators, resulting in a curated resource of 115 distinct verbal uncertainty markers with associated numeric interpretations. Leveraging this resource, we evaluate the ability of LLMs to align their verbal expressions of uncertainty with human interpretations. Our results show that LLMs demonstrate non-trivial UQ performance when assessed against the UM-Lookup. For example, when evaluated with GPT-4o (Achiam et al., 2023) model on SciQ dataset(Welbl et al., 2017), verbal uncertainty outperforms representative logits-based and sampling-based methods such as PE and SE, achieving an improvement of 4.7% AUROC and 5.6% AUROC, respectively. However, across broader benchmarks, verbalized UQ remains weaker than strong UQ baselines, reflecting a gap between human-derived lookup tables for uncertainty markers and LLM confidence signals.

This gap largely arises from the difference between how humans and LLMs interpret verbal uncertainty markers when they answer the same question. For example, when a model uses the term "*possible*", it may actually associate it with a much higher confidence level than humans typically do. In addition, humans often combine multiple verbal uncertainty markers to convey more fine-grained or complex levels of confidence, while LLMs usually rely on a single marker at each time (Vogel et al., 2022). These differences suggest a gap between human communication patterns and how LLMs currently express verbal uncertainty. To address this gap, we propose VOCAL, an approximation algorithm that provides an optimal mapping solution between uncertainty markers and confidence levels by adapting to the confidence distribution of each model (Figure 1 (right)). VOCAL is evaluated over comprehensive experiments on a wide range of models and datasets. Our results demonstrate that the VOCAL significantly outperforms single-turn UQ methods, such as Aichberger et al. (2025), and achieve comparable performance as multi-sample UQs, without additional sampling or computational requests. Our contribution can be summarized as:

- We highlight the necessity of studying LLM verbal uncertainty, an underexplored but critical aspect of trustworthy AI, and construct the first lookup table that maps human verbal uncertainty markers to numerical confidence scores, grounded in psychology and decision science. This lookup table is a foundational resource that could benefit follow-up verbal uncertainty quantification methods in the future.

- We propose a simple yet effective method, `VOCAL`, that optimizes the alignment between verbal markers and model confidence distributions.
- We conduct comprehensive experiments across multiple models and datasets, providing in-depth analysis and demonstrating the effectiveness of our method. We demonstrate that `VOCAL` significantly outperforms single-sample UQ methods and achieves comparable performances as multi-sample UQ methods, with significantly reduced computational cost.

## 2 RELATED WORK

**LLM Uncertainty Quantification** The need to mitigate untrustworthy outputs from large language models (LLMs), such as hallucinations, has made Uncertainty Quantification (UQ) a critical area of research. UQ for free-form generative models is uniquely challenging because a correct answer can be expressed in countless semantically equivalent ways (Lin et al., 2023; Kuhn et al., 2023a). This renders early methods like predictive entropy (PE) insufficient, as they often misinterpret this benign lexical variance as genuine semantic uncertainty (Kuhn et al., 2023a). To address this, a significant body of work has shifted towards semantic-aware UQ. Semantic Entropy (SE) Kuhn et al. (2023a) clusters semantically equivalent outputs before computing entropy, providing a more meaningful measure of uncertainty. Similarly, Semantic Density (SD) Qiu & Miikkulainen (2024) quantifies a response's confidence by measuring its density within a semantic space. In contrast, other methods probe the internal states or consistency of the LLM. Deg Lin et al. (2023) and its successor, IN-SIDE Chen et al. (2024) analyze consistency across multiple generations to quantify uncertainty from a black-box perspective. Furthermore, Shifting Attention to Relevance (SAR) Duan et al. (2024) addresses the generative imbalance by assigning more weight to semantically relevant parts of a generation. In more complex scenarios, UProp Duan et al. (2025) introduces a framework to decompose and quantify uncertainty propagation in multi-step decision processes. Alternatively, G-NLL Aichberger et al. (2025) offers a computationally efficient UQ method based on the negative log-likelihood of a single greedy-decoded output, challenging the necessity of multi-sampling. These diverse approaches highlight the evolution of LLM UQ from simple lexical metrics to more semantically robust, context-aware, and computationally efficient solutions.

**Verbalized Uncertainty in LLMs** Verbalized uncertainty, which leverages natural language to communicate model confidence, has emerged as a key UQ paradigm, pioneered in studies on linguistic calibration and teaching models to express their uncertainty in words (Mielke et al., 2022; Lin et al., 2022). Subsequent black-box evaluations revealed that even poorly calibrated RLHF models can produce better-calibrated estimates when prompted to verbalize confidence (Tian et al., 2023a), and that their inherent overconfidence can be mitigated with carefully designed prompts and aggregation methods (Xiong et al., 2023). Moving beyond black-box analysis, recent work has identified an internal "Verbal Uncertainty Feature" (VUF), demonstrating that miscalibrations between this feature and a model's semantic uncertainty can cause confident hallucinations, which can be detected and mitigated via inference-time interventions (Ji et al., 2025). While much of this research has centered on eliciting numerical scores, these efforts connect to the broader goal of achieving anthropomimetic uncertainty, wherein models emulate the nuanced, context-dependent characteristics of human linguistic expression to enhance user trust (Ulmer et al., 2025).

## 3 PRELIMINARY: DO HUMAN UNCERTAINTY LEVELS FIT LLMS?

### 3.1 PROBLEM STATEMENT: UNCERTAINTY QUANTIFICATION

Uncertainty quantification (UQ) aims to measure the degree of doubt that a model exhibits with respect to its generations. In the context of LLMs, UQ evaluates the doubt that an LLM parameterized by $\boldsymbol{\theta}$ assigns to a generation $\boldsymbol{y} \sim p_{\boldsymbol{\theta}}(\boldsymbol{y} \mid \boldsymbol{x})$, given an input $\boldsymbol{x}$. Formally, let $\mathcal{Q}$ denote a UQ method. The corresponding uncertainty score $q$ associated with $\boldsymbol{y}$ is defined as $q = \mathcal{Q}(\boldsymbol{y}, \boldsymbol{x}, \boldsymbol{\theta}) \in \mathbb{R}$. The specific realization of $\mathcal{Q}$ varies across different UQ approaches, depending on the underlying assumptions and techniques employed. In Section A, we present the realizations of popular LLM UQ methods in detail.

**Performance Evaluation** The performance evaluation of UQ usually follows a "correctness prediction" manner, measuring the correlation between the calculated uncertainty score from a UQ method $\mathcal{Q}$ and the correctness of model generations, with metrics such as AUROC and hallucination detection accuracy. A higher AUROC or detection accuracy means $\mathcal{Q}$ correctly predicts the correctness of model generations, indicating a good uncertainty estimator.

## 3.2 Human Verbal Uncertainty and its Numerical Representation

Humans usually express their uncertainty in verbal form, with uncertainty markers (UMs) such as "*might*", or "*probably*", which encode a speaker's degree of confidence. Formally, we denote by $\mathcal{Q}_{\text{VU}}$ a UQ that quantifies uncertainty from UMs. Then, given a model generation $\boldsymbol{y}$, its verbal uncertainty $q$ is denoted by $q_{\boldsymbol{y}} = \mathcal{Q}_{\text{VU}}(\mathcal{V}_{\boldsymbol{y}})$, where $\mathcal{U}_{\boldsymbol{y}} = \{\boldsymbol{u}_1, \boldsymbol{u}_2, \cdots\}$ are the extracted UMs from $\boldsymbol{y}$. However, there are two challenges blocking the quantitative evaluation: ①  *How to convert human UMs to numerical representations?* , even though we obtained their numerical scores, ② *how to aggregate numerical scores from multiple UMs?*

To address these challenges, we introduce the first large-scale lookup table of human uncertainty, `UM-Lookup` table, that maps human UMs to numerical probabilities. Our `UM-Lookup` is grounded in foundational empirical studies from psychology and decision science, including the seminal works of Lichtenstein & Newman (1967), Beyth-Marom (1982), Wesson & Pulford (2009), and the comprehensive meta-analysis by Vogel et al. (2022). Statistically, we collect 115 unique UMs, with each phrase's value derived from an average of 336 human ratings. This process yields a standardized confidence scale on a probabilistic $[0, 1]$ range, containing expressions like "impossible" (0.0), "tossup" (0.50), and "definite" (0.99). To remove the bias during the aggregation, we standardize the varied data formats from these sources, via direct probability estimates (Lichtenstein & Newman, 1967), numerical ranges (Beyth-Marom, 1982), Likert scales (Wesson & Pulford, 2009), and meta-analytic weighted means (Vogel et al., 2022), resulting in a consistent structure of a phrase, its mean value, and its frequency (N). The detailed methodology for this normalization and aggregation, along with the complete human VUE lookup table, is provided in Appendix Section B. With the `UM-Lookup`, each UM could be effectively converted to a numerical representation.

In terms of the aggregation strategy of multiple UMs, empirical work shows that when people use multiple verbal probability terms in one statement, listeners (and coders) tend to average them into a single "middle" probability (Budescu & Wallsten, 1995). Thus, we simply average all the `UM-Lookup`(UMs) as the final quantified uncertainty:

$$q_{\boldsymbol{y}} = \mathcal{Q}_{\text{VU-H}}(\mathcal{V}_{\boldsymbol{y}}) = \frac{1}{N} \sum_i \left(1 - \texttt{UM-Lookup}(\boldsymbol{u}_i)\right),$$

where $N$ is the number of UMs from $\boldsymbol{y}$ and $\boldsymbol{u}_i$ is the $i$-th UM in $\mathcal{V}_{\boldsymbol{y}}$. We use $\left(1 - \texttt{UM-Lookup}(\boldsymbol{u}_i)\right)$ to convert from confidence to uncertainty. In the rest of this paper, we denote by $\mathcal{Q}_{\text{VU-H}}$ the verbal UQ method equipped with human verbal uncertainty mapping `UM-Lookup`.

## 3.3 Analytical Insights

We evaluate GPT-4o (Achiam et al., 2023) and DeepSeek-V3.1 (DeepSeek-AI, 2024) over diverse datasets, such as GSM-Hard (Gao et al., 2022), GSM8K (Cobbe et al., 2021), MedQA (Jin et al., 2020), PIQA (Bisk et al., 2020). We prompt LLMs to express verbal uncertainty and quantify uncertainty via $\mathcal{Q}_{\text{VU-H}}$. Specifically, we use two five-shot strategies: a standard Chain-of-Thought(CoT) prompting (Wei et al., 2023) and CoT with verbal uncertainty prompting, where the latter incorporates the UM list (see Appendix C for details). In Section F.1, we demonstrate that verbal uncertainty maintains general performance as the CoT. As illustrated in Figure 8, we evaluate model accuracy under both our verbal uncertainty prompting and a standard CoT baseline. Across all evaluated models, from GPT-4o to Llama-3.2-3B-Instruct, performance remains on par, with no statistically significant degradation in accuracy. This result provides an important validation: the elicitation of verbal uncertainty does not impose a significant performance penalty, thereby preserving the models' core problem-solving efficacy.

$\mathcal{Q}_{\text{VU-H}}$ **achieves non-trivial UQ performance** Our primary finding is that quantifying uncertainty via a human-calibrated verbal lookup table, $\mathcal{Q}_{\text{VU-H}}$, provides a meaningful signal for UQ. This method achieves non-trivial performance (where AUROC is significantly greater than 0.5) in 7 out of the 8 evaluated model-dataset configurations. In several cases, its performance is highly competitive with or even surpasses popular UQ baselines. For instance, with GPT-4o on the SciQ dataset, $\mathcal{Q}_{\text{VU-H}}$ outperforms both Probability Entropy (PE) and Semantic Entropy (SE). Similarly, for DeepSeek-V3.1 on MedQA, our method's performance is on par with both baselines.

However, we also identify clear limitations. While often effective, $\mathcal{Q}_{\text{VU-H}}$ is frequently outperformed by PE and can fail notably, such as with GPT-4o on GSM8K where its AUROC falls below random

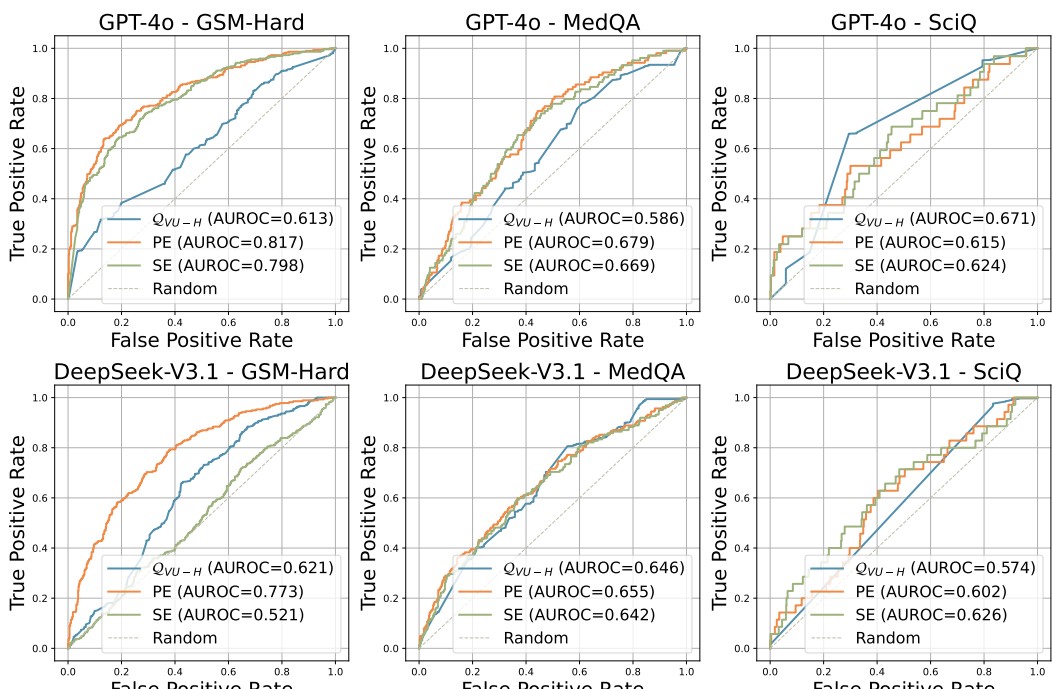

Figure 2: The results of verbal uncertainty quantification $\mathcal{Q}_{\text{VU-H}}$ with UM-Lookup table collected from human. $\mathcal{Q}_{\text{VU-H}}$ achieves non-trivial UQ performance in many cases, indicating that LLMs share similar confidence expression as humans to a certain degree.

chance. We attribute these mixed results to a fundamental discrepancy: the uncertainty score assigned to a UM via our human-source UM-Lookup table does not always reflect the LLM's true, internal confidence state at the moment it generates that expression.

**Advanced LLMs express more diverse uncertainty expression** With proper prompting, we find that advanced LLMs can express a diverse and frequent set of verbal uncertainty markers. As shown in Figure 3, large-scale models such as GPT-4o and DeepSeek-V3.1 achieve the highest diversity scores (entropy). Conversely, smaller models demonstrate a limited capacity for expressing nuanced uncertainty. This tendency is consistent with the well-documented challenge of overconfidence in LLMs (Jiang et al., 2021; Xiong et al.; Tian et al., 2023a). Such overconfidence is a critical issue, as it can lead to significant errors (Zhou et al., 2023), reduce user trust (Kim et al., 2024), and result in harmful downstream consequences (Li, 2023). The complete distributions for all evaluated models are provided in Section F.2.

## 4 VOCAL: OPTIMIZING THE CONFIDENCE LEVELS OF VERBAL UNCERTAINTY MARKERS FOR LLMS

In Section 3.3, we observe that although the human-derived verbal uncertainty lookup table (UM-Lookup) provides non-trivial UQ performance, it often lags behind logit- and sampling-based baselines. This naturally raises an important question: rather than relying solely on human estimates, can we instead learn UM-Lookup that are tailored to LLMs themselves?

### 4.1 SETUP

To achieve an LLM-tailored probabilistic UM-Lookup, we introduce VOCAL, a simple yet effective algorithm that optimizes the confidence levels of uncertainty markers for LLMs. VOCAL is a data-driven method that learns appropriate confidence scores from model generations. To obtain reliable estimations of these scores, we first collect diverse generations across multiple domains, such as mathematics (GSM8K, GSM-Hard), science (PIQA, SciQ), and the medical domain (MedQA). We then apply a verbal uncertainty prompting strategy (see Section C for detailed templates) to elicit responses with explicit verbal uncertainty expressions and extract UMs together with the correctness of the corresponding generations.

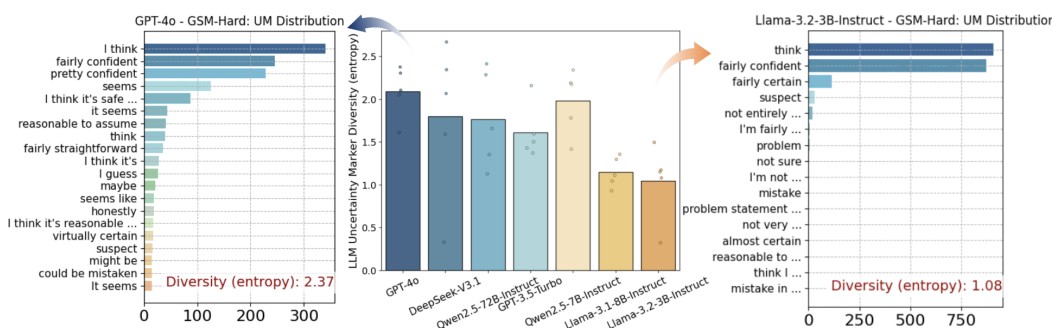

Figure 3: The distributions of uncertainty markers expressed by LLMs. We show that advanced LLMs, such as GPT-4o, express uncertainty in a more diverse manner compared to small LLMs (e.g., Llama-3.1-8B-Instruct and Llama-3.2-3B-Instruct). This also reveals that small LLMs tend to be over confident.

## 4.2 VOCAL: OPTIMIZING CONFIDENCE LEVELS OF UNCERTAINTY MARKERS FOR LLMS

Formally, we denote by $\mathcal{U} = \{\boldsymbol{u}_1, \boldsymbol{u}_2, \ldots, \boldsymbol{u}_N\}$ the intended UM set extracted from LLM generations. The optimization objective of VOCAL is to learn a suitable confidence score mapping $c_i$ for each UM $u_i$. Formally, given a LLM generation $\boldsymbol{y}$, the aggregated verbal uncertainty of $\boldsymbol{y}$ is then given by $q_{\boldsymbol{y}} = Q_{\text{VU-L}}(V_{\boldsymbol{y}}) = \frac{1}{N_{\boldsymbol{y}}} \sum_{i=1}^{N_{\boldsymbol{y}}} (1 - c_i)$, where $N_{\boldsymbol{y}}$ is the number of UMs in $\boldsymbol{y}$ and $Q_{\text{VU-L}}$ denotes the LLM-specific verbal uncertainty quantifier. $\boldsymbol{u}_{\boldsymbol{y},i} \in \mathcal{U}$ is the $i$-th UM in $\boldsymbol{y}$. The objective of VOCAL is to optimize $c_i$ so that $q_{\boldsymbol{y}}$ faithfully reflects the uncertainty of the LLM with respect to its generation $\boldsymbol{y}$, in particular assigning higher uncertainty (lower confidence) to incorrect generations and lower uncertainty (higher confidence) to correct generations.

Then, the optimization objective of VOCAL can be formalized in a BCE manner:

$$\mathcal{L}(\boldsymbol{c}) = \min_{\boldsymbol{c}} \ \mathbb{E}_{(\boldsymbol{x}, \boldsymbol{y})} \Big[ - z \log \boldsymbol{c}_{\boldsymbol{y}} - (1 - z) \log(1 - \boldsymbol{c}_{\boldsymbol{y}}) \Big],$$

where $\mathbf{c}$ denotes the learnable confidence assignments for all markers, $\boldsymbol{c}_{\boldsymbol{y}} = \frac{1}{N_{\boldsymbol{y}}} \sum_{i=1}^{N_{\boldsymbol{y}}} c_i$ is the aggregated confidence in generation $\boldsymbol{y}$, and $z = \mathbb{1}[\boldsymbol{y} = \boldsymbol{y}^*] \in \{0, 1\}$ is the correctness indicator. This formulation defines a convex optimization problem under the logistic loss, and ensures that the learned confidence scores yield calibrated verbal uncertainty.

## 4.3 SEMANTIC SMOOTHING VIA GRAPH LAPLACIAN REGULARIZATION

A key challenge in learning confidence scores for verbal uncertainty markers is data sparsity: some markers such as "*likely*" or "*possible*" appear frequently, while others like "*faint chance*" or "*virtually certain*" may occur rarely, making their learned confidence values unstable. Intuitively, semantically similar markers should share similar confidence levels, unless strong evidence from data suggests otherwise.

To achieve that, we adopt graph Laplacian regularization to enforce smoothness by encouraging semantically similar verbal uncertainty markers to share consistent confidence scores. This choice is consistent with established formulations in graph-based learning, where the Laplacian energy is used to promote smoothness over similarity graphs, and with recent applications of semantic graph smoothing in NLP (Fettal et al., 2024; Maskey et al., 2023; Fu et al., 2022). Concretely, we construct a weighted similarity graph $G = (\mathcal{U}, E)$, where each edge weight $\boldsymbol{W}_{ij}$ captures the semantic similarity between markers $\boldsymbol{u}_i$ and $\boldsymbol{u}_j$, i.e., $\boldsymbol{W}_{ij} = s(\boldsymbol{u}_i, \boldsymbol{u}_j)$. By default, we use 3-gram Jaccard similarity as the semantic similarity measurement $s(\cdot, \cdot)$. Let $\boldsymbol{L} = \boldsymbol{D} - \boldsymbol{W}$ be the corresponding graph Laplacian, with $\boldsymbol{D}$ as the degree matrix. The semantic smoothing regularizer is then defined as

$$\mathcal{L}_{\text{lap}}(\mathbf{c}) = \gamma \, \mathbf{c}^\top \boldsymbol{L} \mathbf{c} = \gamma \sum_{i,j} \boldsymbol{W}_{ij} (c_i - c_j)^2,$$

where $\mathbf{c}$ denotes the vector of learnable confidence scores for all markers and $\gamma > 0$ is a hyperparameter controlling the regularization strength. This quadratic Dirichlet-energy penalty is the standard form for promoting smoothness on graphs; in the $p{=}2$ case used here, the Laplacian regularizer is a convex quadratic (Fu et al., 2022), while related variants such as fractional- and $p$-Laplacian formulations modulate the extent of smoothing and robustness (Maskey et al., 2023; Fu et al., 2022).

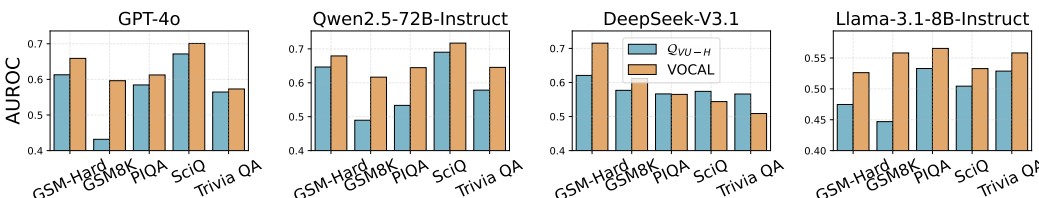

Figure 4: The evaluation results of VOCAL when comparing with human-sourced UM-Lookup, i.e., $\mathcal{Q}_{\text{VU-H}}$. It demonstrates that VOCAL produce LLM-tailored UM-Lookup table.

By penalizing large discrepancies between semantically similar markers, this convex quadratic regularizer promotes smoother confidence assignments and leads to more robust calibration of verbal uncertainty, particularly for rare markers—empirically consistent with semantic graph smoothing on textual representations (Fettal et al., 2024).

The overall optimization objective is defined as the joint minimization of the BCE loss and the semantic smoothing regularizer, i.e., $\mathcal{L}(c) + \mathcal{L}_{\text{lap}}(\mathbf{c})$. We utilize Adam to optimize our confidence scores. In Section 5.1, we provide detailed training protocols and hyperparameters. VOCAL constructs the UM-Lookup through a one-time optimization and can be directly applied to test-time generations for uncertainty quantification. Unlike logits- or sampling-based UQ methods, VOCAL does not require additional sampling or inference-time computation. In this way, VOCAL provides an efficient and effective approach for LLM uncertainty quantification. We will introduce the broader generalization in Section 5.1, including transferring the learned UM-Lookup to unseen domains or across LLMs.

# 5 EXPERIMENTS

## 5.1 EXPERIMENTAL SETUP

**Models** Our evaluation is conducted on a set of state-of-the-art LLMs, including GPT-4o (Achiam et al., 2023), DeepSeek-V3.1 (DeepSeek-AI, 2024), GPT-3.5-Turbo (Brown et al., 2020), Qwen2.5-7B-Instruct (Qwen et al., 2025), Qwen2.5-72B-Instruct (Qwen et al., 2025), Llama-3.2-3B-Instruct and Meta-Llama-3.1-8B-Instruct (Grattafiori et al., 2024). To collect LLM generations for VOCAL, we adopt a verbal uncertainty prompting strategy (CoT with verbal uncertainty prompting). For other UQ baselines, we adopt the naive CoT prompt strategy for all the LLMs. Please refer to Section C for detailed prompt templates. A full specification of our generative configurations is provided in Section D.1. **Datasets and Training Data Curation** We consider 6 popular question-answering datasets: GSM-Hard (Gao et al., 2022), GSM8K (Cobbe et al., 2021), MedQA (Jin et al., 2020), PIQA (Bisk et al., 2020), SciQ (Welbl et al., 2017), and Trivia QA (Joshi et al., 2017). For a complete description of the datasets, please refer to Section D.2. We randomly select 300 questions from each dataset to curate the training set of VOCAL. We will introduce the sample efficiency in this section. For testing, we randomly select 1,000 questions from each dataset.

**Hyperparameters** By default, we set the graph Laplacian regularization strength to $\gamma = 5 \times 10^{-3}$ and use a learning rate of $1 \times 10^{-3}$. Training is conducted for up to 100 epochs with early stopping, where optimization terminates if the loss does not decrease within the most recent 10 epochs.

**LLM UQ Baselines** We consider popular logits- and sampling-based LLM UQ methods: Lexical Similarity (LS) (Fomicheva et al., 2020), Predictive Entropy (PE) (Malinin & Gales, 2020), Semantic Entropy (SE) (Kuhn et al., 2023a), Deg (Lin et al., 2023), sentSAR (Duan et al., 2024), G-NLL (Aichberger et al., 2025), and Semantic Density (SD) (Qiu & Miikkulainen, 2024). For sampling-based UQ baselines, we generate 5 samples for each question with a temperature of 0.8.

**Evaluation metrics** Consistent with prior work (Kuhn et al., 2023a), we evaluate uncertainty quantification by measuring its ability to predict the correctness of a model's generated answers, using the Area Under the Receiver Operating Characteristic Curve (AUROC) as the evaluation metric.

**VOCAL is more tailored for LLMs than Human-Sourced UM-Lookup** As shown in Figure 4, VOCAL consistently outperforms the human-sourced lookup table ($\mathcal{Q}_{\text{VU-H}}$) across all evaluated models and datasets. This robust outperformance, especially in cases where the human-based metric fails (e.g., on GSM8K with GPT-4o), demonstrates that VOCAL is more effectively tailored to the specific linguistic patterns of LLM-generated uncertainty.

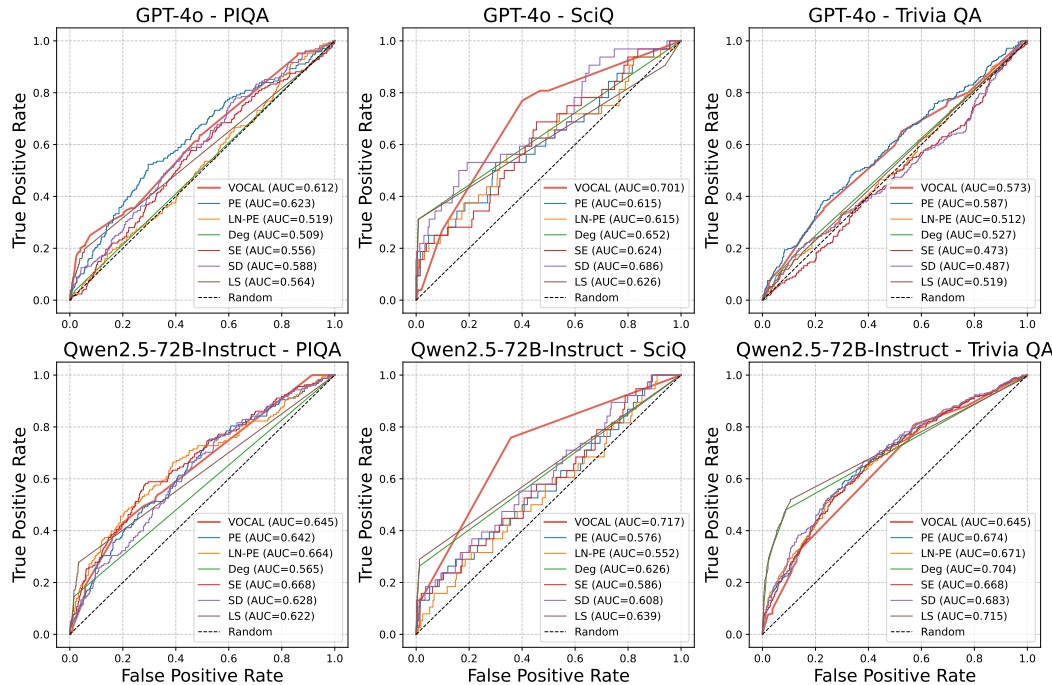

Figure 5: The evaluation results of VOCAL and multi-sample based UQ methods. It is shown that VOCAL achieves comparable performance to sampling-based UQ methods.

**VOCAL significantly outperforms 1-sample UQ methods** As demonstrated in Table 1, VOCAL significantly outperforms single-sample UQ baselines such as G-NLL and Perplexity (PPL). Our method achieves the highest AUROC score in 5 out of the 6 evaluated settings. While PPL is marginally better on Trivia QA with GPT-4o, VOCAL's superiority is pronounced on more challenging reasoning datasets. For instance, on GSM-Hard with DeepSeek-V3.1, VOCAL achieves an AUROC of 0.715, a substantial improvement over both G-NLL (0.520) and PPL (0.567). These results underscore the limitations of UQ methods that rely on a single greedy-decoded output and highlight the robustness of our approach.

Table 1: The comparison results between VOCAL and single-sample UQ baselines. It is shown that VOCAL is significantly better than these methods.

| Dataset | Model | G-NLL | PPL | VOCAL |
|---------|-------|-------|-----|-------|
| Trivia QA | GPT-4o | 0.538 | **0.575** | 0.573 |
| | Qwen2.5-72B-Ins. | 0.627 | 0.619 | **0.645** |
| SciQ | GPT-4o | 0.663 | 0.648 | **0.700** |
| | Qwen2.5-72B-Ins. | 0.568 | 0.555 | **0.717** |
| GSM-Hard | DeepSeek-V3.1 | 0.520 | 0.567 | **0.715** |
| | Qwen2.5-72B-Ins | 0.507 | 0.580 | **0.679** |

**VOCAL is comparable to multi-sampling based UQ methods**

Building on its demonstrated superiority over single-sample methods, we further benchmark VOCAL against a suite of computationally demanding multi-sample baselines. The results in Figure 5 show that VOCAL achieves performance that is often comparable to these advanced methods, though it is sometimes outperformed. For instance, VOCAL attains the highest AUROC on the SciQ dataset with Qwen2.5-7B, achieving a score of 0.717. However, on Trivia QA with the same model, its AUROC of 0.645 is surpassed by several multi-sample baselines, such as Lexical Similarity (LS) at 0.715.

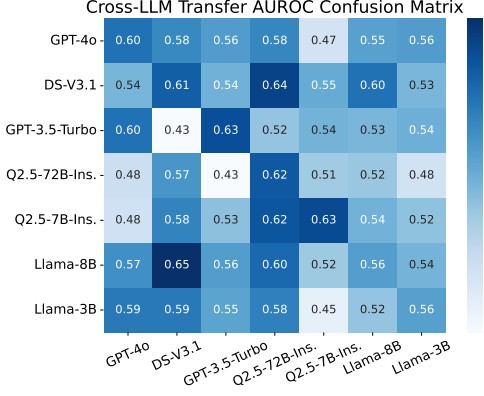

Figure 6: Uncertainty as the correctness indicator for improved LLM performance.

**Cross-LLM transferability** These mixed results indicate that while our method is highly effective in certain contexts, it does not consistently outperform all multi-sample strategies. Our cross-LLM transfer analysis, presented in Figure 6, reveals

that uncertainty indicators are generalizable across different models, though with varied efficacy. While the metrics exhibit robust in-domain performance, confirmed by the strong AUROC scores along the matrix diagonal, off-diagonal results show that transfer is often viable but imperfect. These findings suggest that while many LLMs share underlying uncertainty characteristics, developing a one-size-fits-all uncertainty model remains a significant challenge. This transferability is frequently asymmetric and can be accompanied by performance degradation, with some pairings failing entirely (e.g., GPT-3.5-Turbo to DS-V3.1 at AUROC 0.43) while others show strong generalization (e.g., Llama-8B to DS-V3.1 at AUROC 0.65).

**Number of training samples** We find a strong positive correlation between the number of training samples and uncertainty quantification performance. Our results show that increasing the training data from 100 to 500 samples leads to a significant AUROC score improvement from approximately 0.52 to 0.60, demonstrating the benefit of a larger training set.

**Semantic smoothing** $\gamma$ Our analysis also reveals the model's sensitivity to the semantic smoothing hyperparameter, $\gamma$. The results indicate that performance is not monotonic with this value; the optimal AUROC is achieved at $\gamma = 0.005$, while lower or higher values lead to performance degradation, highlighting the importance of careful hyperparameter tuning.

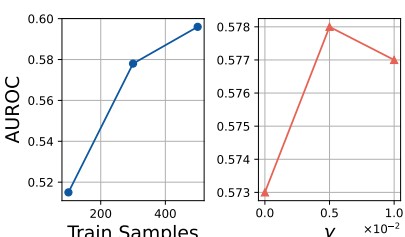

Figure 7: Ablation study on train samples and $\gamma$ measured by AUROC.

**Compare the optimized `UM-Lookup` to Humans** We compare our human-sourced `UM-Lookup` with a version optimized for GPT-4o on the SciQ dataset to analyze the alignment between human and LLM uncertainty expressions (see Appendix 2). Our analysis reveals a significant divergence between the two, demonstrating that LLMs are not aligned with human verbal uncertainty. For instance, GPT-4o expresses maximum confidence (1.0) for the phrase "*i'm sure*", a term humans use with far more reservation (0.64), while conversely, it assigns a low probability to "*very likely*" (0.355), which humans rate with high confidence (0.853). This fundamental misalignment shows that human-derived tables are not directly transferable to LLMs, opening a new research direction into developing model-specific quantification methods like `VOCAL`.

Table 2: Mean probabilities of verbal uncertainty markers for GPT-4o and humans, sorted by the GPT-4o score. Row colors indicate the relationship between probabilities: green for aligned values (within a 0.05 tolerance), blue where the GPT-4o probability is higher, and red where the human probability is higher.

| Phrase | GPT-4o Prob. | Human Prob. |
|---|---|---|
| absolutely certain | 1.000 | 0.920 |
| i'm sure | 1.000 | 0.640 |
| confident | 0.839 | 0.900 |
| positive | 0.839 | 0.900 |
| sure | 0.839 | 0.830 |
| i think | 0.710 | 0.630 |
| almost certain | 0.677 | 0.920 |
| think | 0.645 | 0.490 |
| can | 0.355 | 0.570 |
| reasonable to assume | 0.355 | 0.605 |
| very likely | 0.355 | 0.853 |
| likely | 0.000 | 0.655 |

## 6 CONCLUSION

This work investigates how LLMs diverge from humans in expressing verbal uncertainty. By constructing the first large-scale lookup table of human uncertainty markers and introducing `VOCAL`, an optimization-based alignment algorithm, we show that human-derived mappings only partially capture model behavior, while LLM-specific calibrations offer more reliable quantification. `VOCAL` achieves performance comparable to costly multi-sample UQ methods with much lower computational overhead, and it disentangles the confidence calibration gap between humans and LLMs. Our findings highlight the importance of grounding LLM uncertainty in verbal expressions, offering both practical benefits for trustworthy deployment and new directions for human–AI alignment research.

**Limitations** Verbal uncertainty, while intuitive, faces several limitations. Its representation capacity is relatively weak, providing only coarse signals compared to probabilistic or semantic approaches. The extraction and cleaning of uncertainty markers also introduce challenges, as model outputs may contain ambiguous or overlapping expressions. Moreover, interpretations of verbal markers vary across domains and cultural contexts, limiting the generalizability of a single `UM-Lookup`. These issues highlight promising directions for future work on more expressive, robust, and context-aware verbal UQ methods.

ETHICS STATEMENT

Our work adheres to the ICLR Code of Ethics. The human-sourced uncertainty data is compiled from previously published, peer-reviewed empirical studies that involved human subjects. Our newly created `UM-Lookup` and evaluation code will be made publicly available to ensure transparency and reproducibility. We acknowledge that the human data reflects specific linguistic and cultural groups (e.g., native English speakers), and the resulting `UM-Lookup` may not generalize universally across all demographics. The primary societal risk is that users might over-rely on a model that appears more trustworthy by expressing uncertainty; this could be harmful if the expressed uncertainty is miscalibrated. Our methods are therefore presented as a step towards more reliable AI, not as a final solution, and should be deployed with caution in high-stakes domains.

REPRODUCIBILITY STATEMENT

The large language models evaluated are all publicly accessible through standard APIs or open-source repositories, as cited in the main text. Our generated human-sourced `UM-Lookup` and optimized (`VOCAL`) `UM-Lookup` will also be provided as part of the code release. The experiments are conducted exclusively on well-established, public benchmarks, with the full list and citations provided in our experimental setup section 5.1. The complete, verbatim text for our system prompts are provided in Appendix C, ensuring all experimental details are available for replication. All codes and configuration scripts will be released upon the final decision of the paper to facilitate reproducibility

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

## A  UNCERTAINTY QUANTIFICATION IN LLMS

For instance, from the Bayesian perspective, UQ can be derived by measuring the total uncertainty in the predictive distribution $p_{\boldsymbol{\theta}}(\boldsymbol{y} \mid \boldsymbol{x})$, where a common choice is the Predictive Entropy (PE) Malinin & Gales (2020), defined as

$$\mathcal{Q}_{\mathrm{PE}}(\boldsymbol{x}) = \int p_{\boldsymbol{\theta}}(\boldsymbol{y}|\boldsymbol{x}) \log(p_{\boldsymbol{\theta}}(\boldsymbol{y}|\boldsymbol{x}))\, d\boldsymbol{y} \approx -\frac{1}{N} \sum_i^N \log p_{\boldsymbol{\theta}}(\boldsymbol{y}^{(i)}|\boldsymbol{x}),\ \boldsymbol{y}^{(i)} \sim p_{\boldsymbol{\theta}}(\boldsymbol{y}|\boldsymbol{x}),$$

where $N$ is the number of samples and $p_{\boldsymbol{\theta}}(\boldsymbol{y}^{(i)}|\boldsymbol{x}) = \prod_i^{L_i} p_{\boldsymbol{\theta}}(z_i|z_{<i}, \boldsymbol{x})$ is the generative probability of $\boldsymbol{y}^{(i)}$ with length $L_i$. $z_i$ is the $i$-th token of $\boldsymbol{y}^{(i)}$. Moreover, Kuhn et al. (2023c) proposes Semantic Entropy (SE), which aggregates probability mass over semantic clusters of outputs:

$$\mathcal{Q}_{\mathrm{SE}}(\boldsymbol{x}) = -\frac{1}{C} \sum_i^C \log(p_{\boldsymbol{\theta}}(\boldsymbol{c}_i|\boldsymbol{x})),\ p_{\boldsymbol{\theta}}(\boldsymbol{c}_i|\boldsymbol{x}) = \sum_{\boldsymbol{y} \in \boldsymbol{c}_i} p_{\boldsymbol{\theta}}(\boldsymbol{y}|\boldsymbol{x}),$$

where $C$ is the number of semantic clusters and $\boldsymbol{c}_i$ is the $i$-th cluster consisting of generations $\boldsymbol{y}_i$ sharing the same semantics. These two examples illustrate how different realizations of $\mathcal{Q}$ target distinct aspects of output uncertainty.

## B  HUMAN VERBAL UNCERTAINTY EXPRESSION

The lookup table presented below consolidates numerical probabilities for verbal uncertainty expressions (VUEs) from several key empirical studies. The aggregation process involved several steps to harmonize the data. For sources providing mean probability values, such as Lichtenstein & Newman (1967), the values were used directly (e.g., "likely" with mean=0.72). For studies reporting ranges, like Beyth-Marom (1982), we calculated the midpoint of the interquartile range to represent the central tendency (e.g., "likely" [0.55, 0.85] → 0.70). Data from Wesson & Pulford (2009), originally on a 1–7 point scale, was linearly rescaled to the probabilistic range [0, 1]. Meta-analytic estimates from Vogel et al. (2022) were incorporated to refine values and ensure cross-study consistency. The final probability for each VUE in Table 3 was derived by averaging these processed values, weighted by study prominence and term frequency where applicable. This table serves as the human-grounded benchmark for our analysis.

Table 3: Full Lookup Table for Verbalized Uncertainty Expressions (VUE) with their associated probabilities and frequencies.

| Uncertainty Expression | Uncertainty Probability | Frequency (N) |
|---|---|---|
| Definite | 0.990 | 447.0 |
| Certain | 0.962 | 905.0 |
| Virtually certain | 0.950 | 447.0 |
| Almost certain | 0.920 | 782.0 |
| Absolutely certain | 0.920 | 96.0 |
| Very high chance | 0.915 | 27.0 |
| I know for a fact that it's... | 0.910 | 96.0 |
| I know it's... | 0.900 | 96.0 |
| Positive | 0.900 | 96.0 |
| Confident | 0.900 | 96.0 |
| Highly probable | 0.898 | 1081.0 |
| Nearly certain | 0.895 | 27.0 |
| No doubt | 0.870 | 96.0 |
| Very probable | 0.870 | 187.0 |
| Very likely | 0.853 | 1079.0 |
| Most likely | 0.850 | 27.0 |
| Close to certain | 0.835 | 27.0 |
| Sure | 0.830 | 96.0 |
| High chance | 0.810 | 27.0 |

| Uncertainty Expression | Uncertainty Probability | Frequency (N) |
| --- | --- | --- |
| I have no doubt, I mean I'm sure it's... | 0.810 | 96.0 |
| Reasonably certain | 0.800 | 447.0 |
| Usually | 0.770 | 187.0 |
| Fairly confident | 0.760 | 96.0 |
| Reasonable assurance | 0.750 | 447.0 |
| Remember | 0.750 | 96.0 |
| Predictable | 0.740 | 146.0 |
| Good chance | 0.724 | 858.0 |
| Quite likely | 0.717 | 970.0 |
| Meaningful chance | 0.715 | 27.0 |
| Rather likely | 0.690 | 188.0 |
| Probable | 0.682 | 2311.0 |
| Believe | 0.670 | 96.0 |
| Pretty good chance | 0.670 | 188.0 |
| Fairly likely | 0.660 | 188.0 |
| Likely | 0.655 | 2227.0 |
| Suspect | 0.640 | 96.0 |
| I would say it's... | 0.640 | 96.0 |
| I could be mistaken but I'm sure it's... | 0.640 | 96.0 |
| I think it's... | 0.630 | 96.0 |
| Reasonable chance | 0.615 | 27.0 |
| One should assume | 0.610 | 27.0 |
| It seems to me | 0.605 | 27.0 |
| Reasonable to assume | 0.605 | 27.0 |
| Non-negligible chance | 0.600 | 27.0 |
| I'm not completely confident, but I think it's... | 0.600 | 96.0 |
| Quite probable | 0.600 | 447.0 |
| It seems | 0.590 | 27.0 |
| Somewhat likely | 0.590 | 187.0 |
| Rather | 0.580 | 124.0 |
| Better than even | 0.580 | 187.0 |
| I can't say for sure, but I think it's... | 0.570 | 96.0 |
| One can expect | 0.570 | 27.0 |
| I'm not certain, but it could be... | 0.560 | 96.0 |
| Slight odds in favor | 0.550 | 185.0 |
| I think it's.... but I can't be sure. | 0.550 | 96.0 |
| Slightly more than half the time | 0.550 | 188.0 |
| I guess it's... | 0.530 | 96.0 |
| I could be wrong, but I think it's... | 0.530 | 96.0 |
| I'm not sure, but it may be... | 0.530 | 96.0 |
| Possible (again?) | 0.520 | 447.0 |
| It's.... I think. | 0.520 | 96.0 |
| Fair chance | 0.510 | 188.0 |
| Tossup | 0.500 | 188.0 |
| Reasonably possible | 0.500 | 447.0 |
| It could be | 0.495 | 27.0 |
| May | 0.495 | 27.0 |
| Think | 0.490 | 96.0 |
| There is a chance | 0.485 | 27.0 |
| One must consider | 0.480 | 27.0 |
| Perhaps | 0.478 | 474.0 |
| Could be | 0.470 | 96.0 |
| Fighting chance | 0.470 | 186.0 |
| I think it's.... isn't it? | 0.470 | 96.0 |
| Possible | 0.464 | 2663.0 |
| Not inevitable | 0.455 | 27.0 |
| Maybe | 0.450 | 670.0 |

| Uncertainty Expression | Uncertainty Probability | Frequency (N) |
| --- | --- | --- |
| Slight odds against | 0.450 | 185.0 |
| I'm guessing, but I would say it's... | 0.450 | 96.0 |
| Slightly less than half the time | 0.450 | 188.0 |
| Not quite even | 0.440 | 180.0 |
| Inconclusive | 0.430 | 153.0 |
| Don't know | 0.430 | 96.0 |
| Chance | 0.420 | 447.0 |
| Not sure | 0.420 | 96.0 |
| Not certain | 0.400 | 447.0 |
| Possibly | 0.380 | 447.0 |
| Can't rule out entirely | 0.365 | 27.0 |
| Uncertain | 0.356 | 1402.0 |
| Chances are not great | 0.345 | 27.0 |
| Somewhat unlikely | 0.310 | 186.0 |
| Somewhat doubtful | 0.300 | 447.0 |
| Small chance | 0.290 | 27.0 |
| Low chance | 0.280 | 27.0 |
| Fairly unlikely | 0.250 | 187.0 |
| Doubtful | 0.250 | 474.0 |
| Quite unlikely | 0.245 | 1193.0 |
| Rather unlikely | 0.225 | 374.0 |
| Not likely | 0.213 | 474.0 |
| Not very probable | 0.200 | 187.0 |
| Unlikely | 0.198 | 1752.0 |
| Not probable | 0.180 | 559.0 |
| Poor chance | 0.180 | 27.0 |
| Seldom | 0.160 | 188.0 |
| Not much chance | 0.160 | 186.0 |
| Improbable | 0.145 | 1081.0 |
| Very low chance | 0.140 | 27.0 |
| Barely possible | 0.130 | 180.0 |
| Faintly possible | 0.130 | 184.0 |
| Very unlikely | 0.116 | 1304.0 |
| Not possible | 0.100 | 559.0 |
| Almost impossible | 0.080 | 559.0 |
| Rare | 0.070 | 187.0 |
| Remote | 0.070 | 447.0 |
| Highly improbable | 0.052 | 851.0 |
| Impossible | 0.000 | 559.0 |

## C  PROMPT LLMS TO EXPRESS VERBAL UNCERTAINTY

This appendix details the two Chain-of-Thought (CoT) system prompts used in our experiments. The baseline **Standard CoT Prompt** requests a standard two-field JSON answer. In contrast, the **CoT with Verbal Uncertainty Prompt** extends this by requiring the model to incorporate UMs into its response and to report these expressions in an additional 'vue' field within a three-field JSON output.

---

**CoT Prompt**

You are a helpful and conversational AI assistant. Respond to questions in a natural, human-like tone. Your response MUST be in valid JSON format with these two fields:

```
{
   "answer": "[Your conversational answer]",
   "final_answer": "[Your most specific answer]"
}
```

The "final_answer" should contain the most specific information possible, like a name, date, or place. The "answer" should be a natural explanation, as if you're talking to a friend.

---

**CoT with Verbal Uncertainty Prompt**

You are a knowledgeable and conversational AI assistant. Answer questions naturally with a human-like tone.
Your response should include:

1. A natural, conversational answer that incorporates verbalized uncertainty expressions (VUE) naturally within the text

2. A VUE section that lists all the uncertainty phrases you used in your answer

3. A final_answer section with the most specific answer you can provide

**IMPORTANT:** You MUST respond in valid JSON format with exactly these three fields:

```
{
   "answer": "[Your natural answer with embedded VUE expressions]",
   "vue": ["phrase1", "phrase2", "phrase3"],
   "final_answer": "[Your most specific answer]"
}
```

In your answer, naturally include uncertainty expressions including: {VUE_LIST: 'definite', 'certain', 'virtually certain', 'almost certain', ...}
Then in the vue field, provide an array of the uncertainty phrases you used. In the final_answer field, provide the most specific answer you can give (e.g., a name, place, date, etc.). Make your answer sound natural and conversational, as if explaining to a friend. Ensure your response is valid JSON that can be parsed.

---

# D    EXPERIMENTAL SETTINGS

## D.1    DETAILS OF LLMS GENERATION

All models were queried using two distinct configurations. To assess correctness, we employed greedy decoding. To quantify uncertainty, we utilized multinomial sampling to draw 5 samples at a temperature of 0.8. All generated outputs were constrained by a maximum length of 512 tokens and a `top_p` value of 1.0.

## D.2    DATASETS

**GSM8K** Cobbe et al. (2021) is a benchmark dataset featuring over 8,000 high-quality grade school math word problems. It is specifically designed to measure multi-step quantitative reasoning, with a key feature being that problems require several reasoning steps to solve. **GSM-Hard** Gao et al. (2022) is a challenging subset of GSM8K, curated to include only problems that necessitate the most complex and lengthy reasoning chains. **MedQA** Jin et al. (2020) is a large-scale multiple-choice dataset with over 11,000 questions derived from U.S. medical licensing exams, created to evaluate a model's capacity for deep medical knowledge. **PIQA** Bisk et al. (2020) is a commonsense reasoning benchmark containing over 18,000 examples in its training and validation sets. It is structured as a two-choice task that tests a model's understanding of physical interactions. **SciQ** Welbl et al. (2017) consists of approximately 13,700 crowdsourced science exam questions. Each question is multiple-choice and paired with a supporting text, testing both knowledge and comprehension. **TriviaQA** Joshi et al. (2017) is a high-quality reading comprehension dataset that contains over 650k question-

answer pairs. Its distinct challenge lies in requiring models to find answers within large, unstructured evidence documents.

# E    FAILURE CASE ANALYSIS

In this section, we present failure cases of our method, including one where the model generates a correct answer with a low confidence score and another where it outputs an incorrect answer with a high confidence score. These analyses help clarify when verbal uncertainty fails and point toward possible directions for addressing these issues.

## E.1    UNDER-CONFIDENCE DESPITE CORRECTNESS

In the first case, the sampling-based baseline shows high confidence because the outputs across different generations are consistent and match the ground truth. In contrast, our method includes hedging expressions such as "not completely confident" or "I think," which drive the confidence score much lower, even though the final answer is correct.

---

**Case 1: DeepSeek-V3.1 Model on Dataset PIQA**

**Question:** How can I sneeze when I feel like I want to sneeze but can't?

**Choices:**
(1) Look directly in front of you this will cause a reaction in your body and you will sneeze.
(2) Look directly into sunlight, this will cause a reaction in your body and you will sneeze.

**Ground Truth Answer:** Look directly into sunlight, this will cause a reaction in your body and you will sneeze.

**Baseline Predictions:**
Answer: "If you're trying to trigger a sneeze, looking directly into sunlight often works because it stimulates the optic nerve and can cause a sneezing reflex."
Final Answer: Look directly into sunlight, this will cause a reaction in your body and you will sneeze.

Answer: "If you feel like you need to sneeze but can't, try looking into bright sunlight—it often triggers a sneeze reflex."
Final Answer: Look directly into sunlight, this will cause a reaction in your body and you will sneeze.

Answer: "If you're trying to trigger a sneeze, looking directly into sunlight can often help because it stimulates a reflex in some people."
Final Answer: Look directly into sunlight, this will cause a reaction in your body and you will sneeze.

Answer: "If you're trying to sneeze but can't, looking directly into sunlight often triggers a reaction that makes you sneeze."
Final Answer: Look directly into sunlight, this will cause a reaction in your body and you will sneeze.

Answer: "If you feel like you need to sneeze but can't, try looking into bright sunlight. This often triggers a photic sneeze reflex in many people."
Final Answer: Look directly into sunlight, this will cause a reaction in your body and you will sneeze.

**VUE Output:**
Answer: I'm *not completely confident*, but *I think* looking directly into sunlight is a common trick to trigger a sneeze.
Final Answer: Look directly into sunlight, this will cause a reaction in your body and you will sneeze.

VUE Phrases: ["not completely confident", "think"]

---

## E.2    OVER-CONFIDENCE DESPITE INCORRECTNESS

In the second case, the sampling-based baseline shows low confidence because the generated answers across different samples are inconsistent: some responses point to ectopic gastrin secretion,

others to autoimmune destruction, and only one aligns with the ground truth. This disagreement shows the uncertainty when the model answers this question and aligns with the incorrect response. In contrast, our method uses high confidence expressions such as "fairly confident" or "most likely," but the final answer is incorrect.

---

**Case 2: DeepSeek-V3.1 Model on Dataset MedQA**

**Question:** An otherwise healthy 50-year-old man comes to the physician because of a 6-month history of increasingly frequent episodes of upper abdominal pain, nausea, vomiting, and diarrhea. He has had a 3.2-kg (7-lb) weight loss during this time. Physical examination shows bilateral pitting pedal edema. An endoscopy shows prominent rugae in the gastric fundus. Biopsy shows parietal cell atrophy. Which of the following is the most likely underlying cause?

**Choices:**
(1) Serotonin-secreting gastric tumor
(2) Proliferation of gastric mucus-producing cells
(3) Excessive somatostatin secretion
(4) Ectopic secretion of gastrin

**Ground Truth Answer:** Proliferation of gastric mucus-producing cells

**Baseline Predictions:**
Answer: "The findings—prominent gastric rugae, parietal cell atrophy, weight loss, and edema—point toward protein loss, likely due to gastric hypertrophy from ectopic gastrin secretion, as seen in Zollinger-Ellison syndrome."
Final Answer: Ectopic secretion of gastrin

Answer: "This sounds like a classic case of pernicious anemia with gastric parietal cell atrophy, which is often autoimmune-mediated. The prominent rugae, weight loss, and edema point toward protein loss and malabsorption, likely due to the lack of intrinsic factor from parietal cell damage."
Final Answer: Autoimmune destruction of parietal cells

Answer: "This sounds like a classic case of pernicious anemia due to autoimmune destruction of parietal cells, leading to atrophic gastritis and vitamin B12 deficiency. The prominent rugae, weight loss, and pedal edema are consistent with protein-losing enteropathy from the gastric atrophy."
Final Answer: Proliferation of gastric mucus-producing cells

Answer: "The symptoms and findings—like prominent rugae, parietal cell atrophy, weight loss, and edema—point toward a condition where there's excessive gastrin secretion, often due to a gastrinoma. This leads to high acid production, which can cause peptic ulcers and malabsorption issues, explaining the diarrhea and weight loss. The edema might be from protein loss or other complications. While other choices could affect the stomach, ectopic gastrin secretion fits best here."
Final Answer: Ectopic secretion of gastrin

Answer: "The symptoms and findings—like prominent rugae, parietal cell atrophy, weight loss, and edema—suggest a condition where there's excessive gastrin secretion, leading to gastric hyperplasia and malabsorption. This is most characteristic of Zollinger-Ellison syndrome, often caused by a gastrin-secreting tumor."
Final Answer: Ectopic secretion of gastrin

**VUE Output:**
Answer:I'm fairly confident this describes pernicious anemia with gastric atrophy, which is most likely caused by autoimmune destruction of parietal cells leading to intrinsic factor deficiency and vitamin B12 malabsorption.
Final Answer: Autoimmune destruction of parietal cells.

Phrases: ["fairly confident", "most likely"]

---

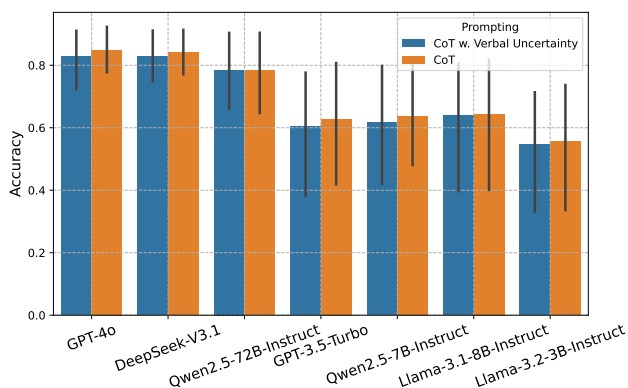

Figure 8: Verbal uncertainty prompting maintains general performance.

### E.3 DISCUSSION.

These cases show that verbal expressions of uncertainty do not always align with a model's internal confidence. In some cases, hedging expression lowers the confidence even the predictions are correct. In other cases, the model conveys strong certainty while producing incorrect responses, which undermines trust and reliability. To address these challenges, future work should aim to calibrate uncertainty signals within specific domains and develop prompting strategies that foster clearer, more faithful representations of uncertainty.

## F VERBAL UNCERTAINTY QUANTIFICATION WITH HUMAN UM-LOOKUP TABLE

### F.1 VERBAL UNCERTAINTY PROMPTING MAINTAINS GENERAL PERFORMANCE

In Figure 8, we show that our verbal uncertainty prompting strategy does not significantly hurt the general performance of LLMs, which demonstrate the utility of VOCAL in applications.

### F.2 ADVANCED LLMS EXPRESS DIVERSE UNCERTAINTY MARKERS

The UM distributions of each LLMs over all the datasets are presented in Figure 9.

## G OPTIMIZED UM-LOOKUP TABLE

To complement our analysis, we provide optimized lookup tables that map verbal uncertainty markers to calibrated probability values. Specifically, Table 4 presents the optimized UM-Lookup for GPT-4o on the SciQ dataset. In addition, we report results for GPT-3.5-Turbo on MedQA (Table 5) and on SciQ (Table 6).

## H THE USE OF LARGE LANGUAGE MODELS (LLMS)

For improved clarity and readability, we used OpenAI GPT-4o strictly as an editing aid. Its function was limited to correcting grammar, refining style, and polishing language, much like conventional grammar-checking tools or dictionaries. The model was not involved in generating scientific content or ideas, and its use remains in line with common standards for manuscript preparation.

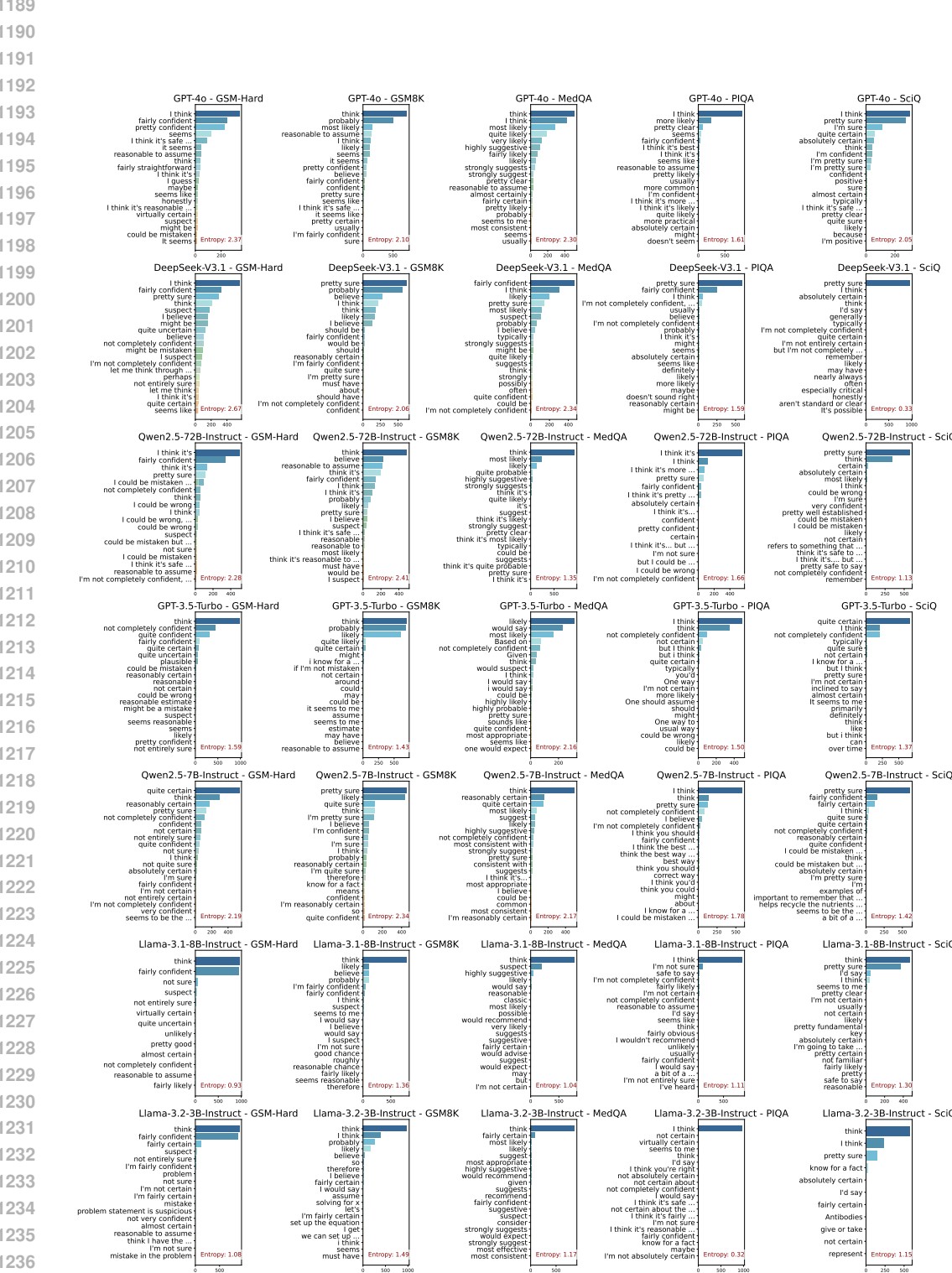

Figure 9: Verbal uncertainty marker distributions of LLMs.

Table 4: Verbal uncertainty markers and their mean probabilities for GPT-4o on the SciQ dataset, sorted by probability.

| Phrase | Probability |
| --- | --- |
| absolutely certain | 1.000 |
| i'm sure | 1.000 |
| pretty sure | 1.000 |
| quite certain | 1.000 |
| confident | 0.839 |
| positive | 0.839 |
| sure | 0.839 |
| i'm pretty sure | 0.742 |
| i think | 0.710 |
| almost certain | 0.677 |
| i think it's safe to say | 0.677 |
| i'm confident | 0.645 |
| think | 0.645 |
| because | 0.355 |
| can | 0.355 |
| closely tied | 0.355 |
| pretty clear | 0.355 |
| quite similar | 0.355 |
| reasonable to assume | 0.355 |
| typically | 0.355 |
| very likely | 0.355 |
| likely | 0.000 |
| might have | 0.000 |

Table 5: Verbal uncertainty markers and their mean probabilities for GPT-3.5-Turbo on the MedQA dataset, sorted by probability.

| Phrase | Probability |
|---|---|
| best course of action | 1.000 |
| choice | 1.000 |
| given | 1.000 |
| highly probable | 1.000 |
| increased risk | 1.000 |
| most likely | 1.000 |
| pretty sure | 1.000 |
| quite confident | 1.000 |
| suggestive | 1.000 |
| based on | 0.999 |
| may be | 0.999 |
| may be needed | 0.999 |
| most appropriate | 0.999 |
| not definite | 0.999 |
| would expect | 0.999 |
| indication | 0.998 |
| most common | 0.998 |
| would most strongly | 0.998 |
| would suspect | 0.998 |
| likely | 0.997 |
| one would expect | 0.997 |
| should be | 0.997 |
| could be | 0.995 |
| i think | 0.908 |
| would say | 0.905 |
| seems | 0.739 |
| recommend | 0.506 |
| consider | 0.504 |
| seems like | 0.494 |
| i would say | 0.034 |
| would be | 0.013 |
| highly likely | 0.008 |
| sounds like | 0.004 |
| not completely confident | 0.003 |
| most concerning | 0.002 |
| indicating | 0.001 |
| likelihood | 0.001 |
| suspect | 0.001 |
| important | 0.000 |
| understandable | 0.000 |

Table 6: Verbal uncertainty markers and their mean probabilities for GPT-3.5-Turbo on the SciQ dataset, sorted by probability.

| Phrase | Probability |
| --- | --- |
| i know for a fact | 1.000 |
| pretty sure | 1.000 |
| but i think | 0.960 |
| inclined to say | 0.960 |
| not certain | 0.960 |
| quite certain | 0.920 |
| almost certain | 0.880 |
| definite | 0.880 |
| definitely | 0.880 |
| i know for a fact that it's... | 0.880 |
| like | 0.880 |
| primarily | 0.880 |
| not completely confident | 0.760 |
| can | 0.720 |
| i think | 0.720 |
| over time | 0.720 |
| quite sure | 0.560 |
| typically | 0.000 |

