# OpenReview forum: "Calibrating the Voice of Doubt: How LLMs Diverge from Humans in Verbal Uncertainty"
_ICLR.cc/2026/Conference — ICLR 2026 Conference Withdrawn Submission_

### Official Review · Reviewer_1Q8T · 2025-10-28

**Soundness:** 2
**Presentation:** 3
**Contribution:** 2
**Rating:** 4
**Confidence:** 3

**Summary:**

In this work, the authors focus on the uncertainty quantification of LLMs using verbalized confidence. They introduce a UM-Lookup dataset to map human verbalized uncertainty markers to numerical values. While simply using this mapping to obtain uncertainty quantification from LLM generations may not be effective, the authors propose VOCAL to learn LLM-tailored mappings. Empirical results have shown the effectiveness of this method with comparison to baselines.

**Strengths:**

1. The proposed dataset UM-Lookup is valuable for uncertainty quantification research.
2. The proposed method VOCAL is novel and efficient in quantifying LLM uncertainty.

**Weaknesses:**

1. As mentioned in Sec. 5.1, the training set and the test set are both randomly sampled from each dataset. It is unclear whether overlap exists between the training and testing samples, which could potentially cause data leakage. Can the authors clarify the sampling protocol with more details?
2. Under the current experimental setup, the authors train VOCAL on 6 datasets and evaluate on the same datasets. This setup does not test clear generalizability of VOCAL. Can the authors clarify whether VOCAL can generalize to unseen datasets beyond the training datasets used for optimization?
3. This paper assumes a correlation between the uncertainty and the correctness of LLM generations. In reality, this might not always hold: the LLM can be overconfident in its incorrect response due to hallucination or other reasons; it can also be less confident in a correct response, as shown in Appendix E. How often does this happen? Such miscalibrated uncertainty may limit the practical application of this work.

**Questions:**

1. The mappings of verbal uncertainty markers to probabilities for LLMs (Tables 4-6) are more extreme than the mappings for humans (Table 3): There exist many moderate values for humans, while the LLM-optimized ones are often polarized toward 0 or 1. This may be because of the preprocessing step in Sec. 3.2, where the authors simply average the ranged probability values. Is this a good practice? Will this lead to any loss of variance information?

---

### Official Review · Reviewer_eTVy · 2025-10-31

**Soundness:** 2
**Presentation:** 1
**Contribution:** 2
**Rating:** 2
**Confidence:** 4

**Summary:**

The authors map commonly used uncertainty phrases to probabilities and demonstrate that detecting these phrases in LLM outputs yields higher than random-chance performance. They propose VOCAL, an algorithm which learns to map LLM-generated phrases to calibrated probabilities.

**Strengths:**

1. The semantic smoothing is an interesting contribution and is a useful addition to the method.
2. Investigating how to interpret LLM-verbalized confidences is an interesting problem.

**Weaknesses:**

Broadly, the paper does not situate itself in relevant literature well, and could be improved through more comprehensive literature review. The conclusions are not always well-supported by the results.
1. Several claims of novelty overreach– for instance, mapping probabilities to verbalized confidences is actually very well studied (see https://pubmed.ncbi.nlm.nih.gov/36073483/ for examples), and thus the claim in lines 78-80 that this is “the first” mapping is untrue.
2. This work appears to suggest that this is the first case of studying linguistic calibration, which is discussed in papers such as https://aclanthology.org/2022.tacl-1.50.pdf and https://arxiv.org/abs/2404.00474. I would have expected to see a section in related work on linguistic calibration, and its omission combined with the statement that this work “opens a new perspective on LLM UQ by grounding it in the verbal dimension” feels misleading.
3. Figure 2 claims to indicate “that LLMs share similar confidence expressions to humans to a certain degree”, while none of the data in Figure 2 seems to be used to compare to humans, but rather to other machine-generated baselines.
4. For an uncertainty quantification methods paper, there is a lack of diverse baselines– for instance, comparisons to other prompting methods, as this scheme falls under the broad heading of prompting methods. This, in turn, makes the less convincing results (such as Figure 6, where many scores are barely above random chance even on in-domain data) harder to interpret positively.
5. Many of the graphs are difficult to read due to small font size, legends covering part of the bar, and misaligned axes.

**Questions:**

The stated motivation for this study is to include more natural descriptions of probabilities in text, rather than probabilities. Would it not be possible to use a better uncertainty quantification method and map these probabilities back to natural language using a table similar to UM-lookup?

---

### Official Review · Reviewer_kiRU · 2025-11-01

**Soundness:** 1
**Presentation:** 2
**Contribution:** 3
**Rating:** 2
**Confidence:** 5

**Summary:**

### Summary

This paper proposes to quantify language model (LM) uncertainty through the use of verbal uncertainty expressions (e.g., “likely”, “unlikely”). To this end, the authors first construct a lookup table to map uncertainty expressions to their numerical representations based on the aggregation of human data from 4 prior user studies in social sciences. Across 2 models and 3 datasets, the average AUROC of the proposed uncertainty method is 61.85 %. Compared to the AUROC of 2 popular baselines, the proposed method falls short of the worst baseline with up to a 18.5% absolute points drop in AUROC.

To improve over the original UQ method the authors propose to calibrate LLM-specific numeric scores associated with each uncertainty expression in the lookup table. The calibration is framed as an optimization problem and includes a regularization term to ensure consistency of confidence scores of semantically similar expressions that seldom occur. Across 2 models and 3 datasets, the average AUROC of the improved uncertainty method is 64.88%. The analysis is further complemented with a study of whether learned calibrated scores for one LLM transfer to a different LLM, with average results per model ranging between 51.57% and 57.29% AUROC.


### Main Contribution

The main contributions of this work are:
-  **the proposed uncertainty quantification method**: the use of uncertainty expressions in generated text is novel in itself; the alignment of the numerical representations using a regularization term to ensure the confidence consistency of rarer but semantically similar expressions is also novel.
- and the **extensive experimentation** that considers up to 7 models, 5 datasets, and up to 6 popular baselines that include both logit-based and sampled-based baselines.

**Strengths:**

S1. The proposal of an uncertainty quantification (UQ) based on a lookup table for the numerical scores associated with verbal uncertainty markers (e.g., “likely”, “unlikely”) is novel.
S2. The calibration of confidence scores for each uncertainty marker and use of regularizer to ensure confidence consistency of semantically similar uncertainty expressions is novel.
S3. A lot of interesting insights and detailed analysis (e.g., ablations on number of training samples, regularization parameters)
S4. Evaluation of 7 models and 5 datasets.

**Weaknesses:**

W1. In its current version, the experimental design and results do not provide enough evidence to support some claims (see Questions > Results Interpretation).
W2. Relevant work is not properly discussed, namely prior work related to the construction of UM-lookup tables and analysis of differences in LLM’s and human’s perceptions of uncertainty markers (See Questions > Contribution-wise)
W3. There are several methodological questions that remain unaddressed, including questions about construction of the table, how the methodology handle uncertainty markers that are not in the lookup table (See Questions > Methodology).
W4. The inconsistency in the reported results makes it difficult to draw conclusions. Some figures report the results for GPT-4o and DeepSeek (Figure 2), others focus on GPT-4o Qwen models (Figure 5). Additionally, several figures lack information about the experimental design (e.g., what datasets and/or models are being used to report the values). Examples of such figures include Figure 6 and 7)
W5. Statistical significance is mentioned but no evidence of actual hypothesis testing or error bars is provided.

I’ve listed the weaknesses based on my understanding of the submitted paper. I’ve thoroughly addressed each weakness in the questions section. It is possible that parts of the text were less clear and I would like to engage in a discussion with the authors to clear any misunderstanding.

**Questions:**

### Questions

**Contribution-wise**:

1. Lines 79-81 claim “we construct the first verbal uncertainty marker lookup table (UM-Lookup) that maps qualitative expressions of uncertainty to numerical representations”. However, smaller versions of such lookup tables exist since (at least) 1964 with [Sherman Kent’s seminal work on words of estimative probability](https://web.archive.org/web/20070613122040/https://www.cia.gov/library/center-for-the-study-of-intelligence/csi-publications/books-and-monographs/sherman-kent-and-the-board-of-national-estimates-collected-essays/6words.html). Other similar lookup tables have since been proposed in various domains and language (e.g., [Fagen-Ulmschneider et al 2019](https://waf.cs.illinois.edu/visualizations/Perception-of-Probability-Words), [Willems et al 2019](https://arxiv.org/abs/1901.09686)). Some of which have effectively been used in prior work to augment LLMs with the ability to communicate using calibrated uncertainty markers  ([Chaudhry et al 2024](https://arxiv.org/abs/2409.12180)). If I understand correctly, **the existence of such prior work invalidates the claim that this work proposes the first “verbal uncertainty marker lookup table”**. Can you please clarify the differences from this work from pre-existing work in social sciences and/or NLP work that uses such mappings to align verbal and numerical uncertainty expressions?
2. Line 100 claims “VOCAL significantly outperforms single-turn UQ methods”. The reported results (in Table 1) only refer to 2 models and 3 datasets, with performance differences varying between -0.009 and 0.148 absolute points w.r.t. to the best competing baseline; avg performance difference in reported values is 0.075 points). Moreover, the results for TriviaQA and SciQ are reported with GPT-4o and Qwen2.5-72B-Ins but DeepSeek-V3.1 is reported for GSM-hard instead of GPT-4o. The proximity of the AUROC values, the lack of standard errors or hypothesis testing, and inconsistency of results raises questions about the actual significance of the claim in Line 100. Can the authors please provide the results across all LMs and datasets and comment on how they determine significance in this case?
2. The paper does a good job in discussing logit-based and sampling-based uncertainty quantification approaches in LLMs. However, it does not mention the recent advances in the use of fine-tuning approaches to calibrate LLMs numerical scores. A few works potentially relevant to this work that focus on the alignment between numerical and verbal uncertainty are [Chaudhry et al 2024](https://arxiv.org/abs/2409.12180) and [Yona et al 2024](https://aclanthology.org/2024.emnlp-main.443).


**Methodology-wise**:

1. **UM-Lookup table construction**: Construction of the lookup table includes aggregating and normalizing the results from 4 different sources. All these sources can introduce noise: numerical assessments are collected based on different variable types (continuous, categorical using Likert Scale, intervals) and uncertainty markers may be studied in isolation (i.e., give the word to a human and ask to rate uncertainty related to that) or in the context of a statement (i.e., give a sentence to the human and ask to rate the uncertainty). Both factors have been shown to greatly impact users’ assessment [Dhami and Mandel 2022](https://www.sciencedirect.com/science/article/pii/S1364661322000602). The information in the appendix B lacks sufficient details about how the numerical assessments were collected from humans (e.g., were they asked to create their own lexicon of uncertainty markers or was there a predefined lexicon that everyone used? Were the uncertainty expressions collected in the context of statements? In which domain?)
2. How does the proposed UQ method work whenever an uncertainty marker is generated  but not included in the lookup table?
3. The proposed uncertainty quantification (UQ) method uses “1-UM-Lookup(u_i) to convert from confidence to uncertainty”. There is a discussion to be had about what “confidence” means. Consider the uncertainty markers “impossible” (p=0.0 in Table 3) and “definite” (p=0.990 in Table 3). In both cases the expressions manifest confidence about the event (i.e., if something is *impossible*, we are certain that the event will not happen, whereas if something is *definite* we are certain that the event will happen). However, under the current approach, “impossible” will be assigned an _uncertainty value_ of 1, which does not consider the nuances we discussed before. I suggest that the authors refrain from using the term “confidence in line 192” and instead use “probability” or to clearly define what they mean by confidence. Did the authors consider other transformations of the numerical representations?
4. The authors mention in Appendix D that two different generation configurations are used: (1) greedy to assess correctness, and (2) sampling with temperature=0.8 and top_p=1.0. If I understand correctly, greedy generations are used only to compute results in Figure 8, and sampling is used for other experimental results. Is this a correct understanding?
       -  If so, it is possible that there is a discrepancy between the accuracy results in Figure 8 and verbalized uncertainty results evaluated in the remainder of the paper. Are uncertainty expressions present in greedy generations? How does the correctness change when using multinomial sampling as opposed to greedy?
5. As part of the Semantic Smoothing via Graph Laplacian Regularization, the authors propose to measure semantic similarity using the 3-gram Jaccard Similarity. Could you please elaborate on how such measure is used to compare expressions that have less than 3 words (for instance, what’s the semantic similarity value of the pairs (“highly likely”, “definite”) and (“I have no doubt, I mean I’m sure it’s”, “Rather”)).
6. The use of a Laplacian regularizer is interesting and intuitive. But how relevant is this term for uncertainty quantification? Figure 7 examines the impact of three regularizer strengths–{0, 0.005, 0.01}--in the AUROC of the VOCAL method. While using regularizer strength of 0.005 seems to yield an improvement over not using it, the reported AUROC improvement is about 0.005 absolute points. Being a very small difference, I wonder about the actual significance of the Laplacian smoothing. Did the authors perform any hypothesis testing to validate its significance? If possible can you add error bars to Figure 7 or report mean and standard deviation over AUROC values obtained from evaluating different datasets?
      - If my interpretation of the results is correct, then **in its current form, the claim “promotes smoother confidence assignments and leads to more robust calibration of verbal uncertainty”** (in lines 335-336) **is not sufficiently supported**, i.e., the experiments do not provide statistically significant support to this hypothesis.

**Results Interpretation**:
1. Lines 204-205 claim no statistical differences between models’ performance when generating answers with different prompts. However, the claim is based on experiments that lack details: which datasets were used to compute such results? How large are these datasets? How was performance evaluated in this case? A paired hypothesis testing should be done to ensure that there are no significant differences at an individual level, which could be obscured by considering averages of correctness per example.
2. Lines 245-246 claim that “advanced LLMs can express a diverse and frequent set of verbal uncertainty markers” and “smaller models demonstrate limited capacity for expressing nuanced uncertainty”. To support these findings, the paper measures the entropy of the distribution over generated uncertainty markers. However, a few observations raise concerns about these claims:
      2.a.) It is unclear whether the entropy measure is comparable across models, since they may generate different sets of uncertainty expressions. If entropy is compared over different supporting sets across models, then the entropy values are not directly comparable across models. Please consider clarifying what’s the support of entropy computation (is it the expressions in the UM look up table?) and what happens if models generate an expression that is not included in such a table, or if they do not generate any uncertainty expression at all (i.e., if counts for all UMs are 0, is entropy equivalent to the entropy of a uniform distribution)?
      2.b) If I understand correctly, higher entropy implies a more diverse set of uncertainty markers, since it implies that the models do not resort to a single uncertainty marker.  However, **entropy does not account for the uncertainty magnitude of the markers used by the LLM**. As a consequence, **it may be the case that most expressions used by advanced LLMs are still conveying overconfidence** (but using a more diverse set of expressions to convey it instead of withholding them). It would be nice if the authors could add some qualitative analysis on this, since in the semantic perspective, expressiveness of uncertainty may not be as diverse as initially measured by the entropy measure. Such a measure is necessary to be able to support the claim “this also reveals that small LLMs tend to be overconfident” (in Figure 3).
      2.c) The second claim, that “smaller models demonstrate limited capacity for expressing nuanced uncertainty” is not obvious to me. Firstly, although the experimental setup considers several models at different model scales, they are mostly from different model families which makes direct comparisons difficult due to differences in training data and architectures. Two pairs of the same model family are Qwen 2.5 Instruct (7B and 72B) and Llama 3.2 3B and Llama 3.1 8B instruct. While it appears that the larger llama model has higher entropy (and thus more expression diversity) the same is not observed between 72B and 7B Qwen models: for which the smaller model exhibits higher diversity. If I’m interpreting the results correctly **this contradicts the original claim that smaller models are less diverse than larger models**.
3. Line 375 claims that “VOCAL consistently outperforms the human-sourced lookup table across all evaluated models and datasets”. These results are presented in Figure 4 and while VOCAL seems to lead to overall improvements across models and datasets (with the exceptions of 3 datasets in DeepSeek model and TriviaQA in GPT-4o model), the  bars are close to each other.  Can the authors comment on the statistical significance per model, dataset configuration?
4. The comparison between VOCAL and single-sample UQ baselines concerns 3 datasets and 2 different models. In GSM-hard a different model is used. Can the authors provide the full set of results across all models and datasets?
5. The paper investigates the impact of the number of training samples in lines 440-446. The reported value for AUROC using 300 training samples (per dataset) is 0.58. According to the experimental setup, VOCAL is fine-tuned using 300 training samples per dataset, so I would expect the value reported in Figure 7 for 300 training samples to match the average AUROC in Figure 5 (~0.649). However this is not the case (perhaps because results in . Can the authors clarify what setting is used to investigate the impact of training examples (e.g., model, training data, evaluation set, and whether the model evaluated for n=300 in Figure 7) is exactly the same used to produce the plots in Figure 5?


**Generalization of the proposed method**:
1. How generalizable is VOCAL to different domains? Are the numbers learned in one domain (e.g., medical) transferable to another domain (e.g., legal)?
2. Overall the AUROC values seem relatively low. The average AUROC values reported for [semantic entropy are approximately 0.8](https://www.nature.com/articles/s41586-024-07421-0) (according to Figure 2 in the original paper). Conversely, the average AUROC reported in Figure 5 of this paper range from 0.473 and 0.668–a set of values that is quite different from the originally reported values. Can the authors comment on the configuration used for semantic entropy? Or perhaps provide an hypothesis for the large performance gap?


### Writing

The paper is overall well written but some inconsistencies make it difficult to parse. Some improvements in text and figures formatting, typos, and reading flow must be integrated to improve the paper before it is ready for camera. I’ve listed some suggestions below:


**Reading flow / Clarity**:

- Lines 47-49: The two uncertainty quantification (UQ) categories are introduced (sample-based, logit-based), followed by the description of one method from each. It is unclear at first why these methods are being described. So, I suggest adding a preposition, e.g.,  “For instance, Malinin and Gales (2020) introduced …”
- Lines 67-70 (Figure 1 caption): The abbreviation UM has not been introduced yet.  Could improve clarity if you define it in caption by expanding it as “uncertainty markers (UM)”.
- Lines 87-90: in the text “verbal uncertainty” is used to describe both original verbalized confidence (that outputs a number) and the proposed method (outputs uncertainty expressions), making the statements in lines 87-90 ambiguous. One suggestion is to use different terminology for both approach types and use them consistently throughout the paper. Specifically, I suggest addressing ambiguity in the sentences:
       - line 87: “verbal uncertainty outperforms representative logits-based and sampling-based methods”.
       - line 89: “verbalized UQ remains weaker than strong UQ baselines”: is this referring to the proposed method
       - note: line 53 defines “verbal uncertainty, where models are asked to output a confidence score (often on a 1-100 scale) in natural language form (Tian et al 2023b).” which overlaps with the terminology used in lines 87 and 89. However, I believe they refer to different methodologies.
- Lines 100-102 discuss the superiority of the proposed method while referring to its sampling efficiency (i.e., requires a single sample). However, the proposed method requires a training set (of at least 300 datapoints per dataset to yield improvements in AUROC superior to random) points per dataset (as indicated in experiments in Figure 7). This should be communicated upfront to be clear about the requirements and limitations of the proposed method.
- Equation 1: There’s no connection between right handside of the equation and the left handside. One suggestion to make the definition clearer is: “$\sum_{i}$” → “$\sum_{u_i \in \mathcal{V}_y}$”.
- Section 3.3 could be greatly improved. The paper begins by stating that GPT-4o and DeepSeek are evaluated giving the impression that only those two models were evaluated. This becomes confusing when later it refers to Llama 3.2-3B-Instruct. Moreover, this section mentions the use of 4 datasets but results are reported for only 3 datasets (in Figure 2).
- Figure 8: Important information related to which dataset was used to conduct such evaluation is missing.
- Lines 211-213 claim “in several cases, its performance is highly competitive with or even surpasses popular UQ baselines”. However, the average AUROC difference to the best baselines in Figure 2 is -0.072, suggesting that the proposed method systematically falls short of the best UQ baseline. The authors might want to consider being quantitative about how often their proposed method surpasses other baselines and provide average improvement. Moreover, they may want to add a comment on why, despite being slightly worse than popular UQ baselines, one may still want to consider the proposed method.
      - While the authors refer to the failure of GPT-4o on GSM8K, I could not find such results in either the main paper or appendix.
- Conclusion in lines 241-243 could be phrased as miscalibration between models’ internal confidence and human confidence. Could also discuss how that hypothesis relates to prior work [Belem et al 2024](https://aclanthology.org/2024.emnlp-main.483/).
- Notation in Section 4.2 is a bit distracting. The paper keeps introducing the set of uncertainty markers but this is rarely mentioned in the equation or objective of the Binary cross-entropy (BCE).
- Line 294 introduces BCE but does not expand what it is. I assumed it refers to the binary cross entropy but suggest the authors expand the term for clarity and/or add a reference to relevant paper.


**Formatting**:

- Line 50: “Kuhn et al (2023b)” → Citation is in the wrong format. Should be in parenthesis.
- Line 86: “SciQ dataset\cite“ → “SciQ dataset \cite”. The lack of whitespace between text and citation happens a few times throughout the paper. Please consider replacing all occurrences.
- Line 117. “LLM Uncertainty Quantification” → “LLM Uncertainty Quantification.” This type of formatting occurs throughout the paper. I suggest adding a “.” at the end of the inline text section headers to clearly distinguish a section header from the remaining of the text. Alternatively, the authors can use the \paragraph instead of \textbf. This is a somewhat subjective preference of formatting.
- Paragraph 1 in Section 2: Citation format (in lines 123, 124, 127, 130, 132) should be in parenthesis.
- Figure 2: X-axis is difficult to read as the model name overlaps each other. One suggestion is to align the x-labels to the right, so that the end of the model name aligns with the tick.
- Figure 4 and 7: add error bars to help draw more meaningful conclusions from the plots.

**Typo**:

- line 101: “single-turn” → “single-sample”
- line 156: “Section” → “Appendix”
- line 166: “$\mathcal{U}_y$” → $\mathcal{V}_y$? There is no U in the definition of $q_y$
- line 168: “representations? ,” →”representations?,”
- line 245: “uncertainty expression” → “uncertainty expressions”
- line 357: “Section D.1.” → “Appendix D.1.\n Datasets and Training Data Curation”
- line 358: “Section D.2.” → “Appendix D.2.”
- line 416: “UQ methods\n Building on its” → “UQ methods. Building on its” (There’s an extra newline character.
- line 902 and lines 915: both mention “possible” but the former has “Possible (again?)” is this a typo?


**Missing Citation**:

- Line 53: Missing citation to [Lin et al 2022: _Teaching Models to Express Their Uncertainty in Words_](https://openreview.net/forum?id=8s8K2UZGTZ).
- Line 72-73: The claim “_people prefer qualitative terms such as “possible,” “likely,” or “almost certain” in daily communication_” should be supported by references in social sciences. Example of relevant work includes [Wallsten et al (1993)](https://link.springer.com/article/10.3758/BF03334162) or [Erev and Cohen 1990](https://www.sciencedirect.com/science/article/pii/074959789090002Q?via%3Dihub).
- Line 91-93: Missing citation to [Belem et al 2024](https://aclanthology.org/2024.emnlp-main.483) which investigates the disparity between how humans and LLMs perceive uncertainty markers.
- Line 94-95: Missing citation to [Zhou et al 2024](https://aclanthology.org/2024.acl-long.198) which finds that only 5% of the LLM-generated answers contains any type of uncertainty marker.
- Line 147: Other relevant references [Kim et al 2024](https://dl.acm.org/doi/10.1145/3630106.3658941), [Steyvers et al 2025](https://www.nature.com/articles/s42256-024-00976-7) further support the idea that appropriate use of uncertainty markers does enhance user trust.

**Missing relevant discussion**:

- Lines 71-72 mention the unnaturalness of expressing confidence through numerical quantities in language form. However, another well known problem of verbalized confidence is their quick saturation, i.e., methods do not produce a diverse set of scores which typically harms their discrimination power (AUROC).


**Missing important details for reproducibility**:
- Details regarding how different baselines are operationalized are missing. For instance, which models were used to operationalize Semantic Entropy? Are default settings used by default?
- Several of the reported evaluation results (e.g., Figures 3, 6, and 7) it is unclear what dataset the results are being reported over. Given the various domain and question formatting differences to be clear about dataset size and which evaluation datasets are being used for evaluation.

---

### Official Review · Reviewer_aLGn · 2025-11-07

**Soundness:** 2
**Presentation:** 2
**Contribution:** 2
**Rating:** 2
**Confidence:** 4

**Summary:**

Authors make following contributions:
C1. They compose large-scale look up table of human-uncertainty.
C2. They propose a post-hoc calibration procedure for verbalized uncertainty markers in LLM outputs. (with special credit to an interesting regularization method dubbed "semantic smoothing" they propose)

**Strengths:**

S1. I find both C1 and C2 valueable contributions, with C2 being more interesting of the two.

**Weaknesses:**

W1. I think the authors miss some important related prior work.

W1a. I think that what the authors propose is essentially "post-hoc calibration" or "post-hoc scaling" (e.g. Platt scaling) but for verbalized uncertainty.
I think it would be worthwhile to relate this contribution to this large previous body of work.

W1b. I think this work should be acknowledged and somehow related to: https://arxiv.org/abs/2404.00474. Could the authors please comment on this work? (or explain why it's irrelevant, if they think I'm wrong)

W1c. other related works: https://arxiv.org/abs/2404.00474, https://arxiv.org/abs/2410.09724, https://arxiv.org/abs/2508.18847, https://arxiv.org/abs/2505.14489 (non-exhaustive list)

W2. When it comes to communicating uncertainty to a user, the quantity you want to measure is calibration (e.g. with expected calibration error, ECE) to allow human decision-making (involving probabilistic reasoning), not AUROC in a selective-prediction/correctness-prediction setting because that's a rank-based metric.
Let me know if this argument is unclear and needs more explanation - happy to do so during the rebuttal period!

W2a. In caption of Fig 3 you say "LLMs tend to be overconfident". It does not follow from the plot immediately. If the model on the right hand side gets close to 100% right, that distribution of verbal-confidence-markers might be in fact well calibrated.
At the end of Sec 4.2 you say "yield calibrated verbal uncertainty", but you don't measure calibration anywhere. Please include calibration plots . I will seriously consider increasing the score if you do this. This might help: https://github.com/apple/ml-calibration. I would say don't even include AUROC evaluations (and hence predictive-entropy and semantic-entropy results) - I don't think that's a good evaluation methodology for what you're trying to achieve, and I don't think PE/SE/LS/Deg/sentSAR/SD are necessarily relevant baselines for your method. (G-NLL might be relevant because you can evaluate its ECE).
I'd suggest authors look through the papers in W1c and choose some relevant baselines. (I haven't evaluated carefully which methods would make for appropriate baselines for your method.)

W3. L463: Seeing that "I'm sure" corresponds to 64% probability (and "sure"=83% in Table 2), in my eyes, casts doubt on the quality of data used to compose UM-Lookup (C1). Individually those probabilities don't seem to line up with my intuitive understanding of those terms. I also cannot understand where such a significant difference between the two comes from? (For me they mean the same.) Doesn't that come across as way off for the authors?

**Questions:**

Questions

Q1. L242 - "does not always reflect the LLM's true internal confidence state". Is there evidence supporting this statement? What is the "true internal confidence state"? Can we measure it?


Remarks

R1. I think I'd find the following a more interesting/practical contribution than C2 (while a downstream application of C1): finetune the LLM to calibrate its use of different verbal-confidence-markers to be in line with probabilities humans imply (as indicated by C1).

R2. I think the authors will find this paper interesting and it should probably be cited in this work because it : https://arxiv.org/abs/2404.00474

R3. "Tian et al., 2023" is doubly defined.

R4. L73: "capture the nuance of human reasoning" -> personally, I think the use of these terms captures the ambiguity of human reasoning about frequency.

R5. Spell out "BCE" the first time you use the abbreviation.

R5. L208 - mention which set of results (Table/Figure?) the reader should look at reading this paragraph.

R6. Abstract: I find the phrase "disentangles the calibration mismatch" confusing and vague.

R7. Some serious vspace hacking around Sec 6 header.

---

### Note · Authors · 2026-01-16

I have read and agree with the venue's withdrawal policy on behalf of myself and my co-authors.